# Application of a Fusion Model Based on Machine Learning in Visibility Prediction

Maochan Zhen [1,2,3], Mingjian Yi [4], Tao Luo [1,2,3,*], Feifei Wang [1,2,3], Kaixuan Yang [1,2,3], Xuebin Ma [1,2,3], Shengcheng Cui [1,2,3] and Xuebin Li [1,2,3]

1 Key Laboratory of Atmospheric Optics, Anhui Institute of Optics and Fine Mechanics, Chinese Academy of Sciences, Hefei 230031, China
2 Science Island Branch of Graduate School, University of Science and Technology of China, Hefei 230026, China
3 Advanced Laser Technology Laboratory of Anhui Province, Hefei 230037, China
4 School of Environment and Energy Engineering, Anhui Jianzhu University, Hefei 230009, China
* Correspondence: luotao@aiofm.ac.cn

**Abstract:** To improve the accuracy of atmospheric visibility (V) prediction based on machine learning in different pollution scenarios, a new atmospheric visibility prediction method based on the stacking fusion model (VSFM) is established in this paper. The new method uses the stacking strategy to fuse two base learners—eXtreme gradient boosting (XGBoost) and light gradient boosting machine (LightGBM)—to optimize prediction accuracy. Furthermore, seasonal feature importance evaluations and feature selection were utilized to optimize prediction accuracy in different seasons with different pollution sources. The new VSFM was applied to 1-year environmental and meteorological data measured in Qingdao, China. Compared to other traditional non-stacking models, the new VSFM improved precision during different seasons, especially in extremely low-visibility scenarios (V< 2 km). The TS score of the VSFM was significantly better than that of other models. For extremely low-visibility scenarios, the VSFM had a threat score (TS) of 0.5, while the best performance of other models was less than 0.27. The new method is promising for atmospheric visibility prediction under complex urban pollution conditions. The research results can also improve our understanding of the factors that influence urban visibility.

**Keywords:** visibility prediction; XGBoost; LightGBM; stacking; fusion model

## 1. Introduction

As a high-tech method, remote sensing has been applied to all aspects of human life and production. The transparency of the atmosphere has a significant impact on remote sensing. Atmospheric visibility (V) is an important indicator of atmospheric transparency. It is defined as the maximum distance at which a person with normal vision can recognize an object of a certain size from the background (sky or ground). V is mainly affected by the aerosols, absorption gases, and meteorological factors in the atmosphere. Research on visibility prediction is critical for the atmospheric correction of remote sensing images [1–4].

At present, the prediction methods for visibility include numerical model prediction and statistical prediction based on machine learning. Numerical model prediction is mainly based on aerodynamic theory and physical and chemical processes. It establishes an environmental meteorological numerical model system to simulate pollutants, humidity, liquid water content, and other elements in the atmosphere by using various meteorological data and emission source data. It predicts atmospheric visibility by calculating the contribution of various elements to atmospheric extinction, according to the theory of atmospheric optics [5,6]. Widely used models include community multi-scale air quality (CMAQ) [7,8], developed by the U.S. Environmental Protection Agency; the weather research and forecasting (WRF) model coupled with chemistry (WRF-Chem) [9], jointly developed by the National Atmospheric Research Center and the National Oceanic

and Atmospheric Administration [10]; and the haze numerical prediction model (CMA), developed by Unified Atmospheric Chemistry Environment (CAUCE) [11]. Numerical model prediction requires an in-depth understanding of the physical and chemical mechanisms of detailed regional emissions. However, it is difficult to accurately quantify each atmospheric process theoretically, leading to prediction errors and uncertainty [12–14].

In recent years, with the development of machine learning, many scholars have used XGBoost, LightGBM, random forest (RF), support vector machines (SVMs), and other machine learning algorithms to conduct visibility prediction research [15–27]. Tang et al. [15] proposed a model using XGBoost combined with Markov chain to predict atmospheric visibility in Shenzhen. The experiment was used to train and predict visibility using meteorological parameters as influencing factors. The experiment achieved a good prediction effect. Yu et al. [16] predicted visibility in Beijing using LightGBM combined with meteorological parameters and $PM_{2.5}$ concentration. The experimental results show that the prediction effect of LightGBM is good and close to the observed value of visibility. The work of Zhang et al. [27] shows that XGBoost and LightGBM are the most advanced regression models.

Studies on visibility prediction usually use a single machine learning model and historical meteorological data to determine the relationship between visibility and other observations [15–26]. However, in addition to meteorological conditions, visibility is also affected by factors such as pollutants and aerosol chemical composition [22]. Especially in China, the sources of urban pollution are extremely complex [24]. Methods based on a single model and limited meteorological factors often have difficulty producing accurate predictions in some scenarios, especially in low-visibility conditions.

The fusion model [25], also known as ensemble learning, is one of the most popular research directions in the field of machine learning. The basic idea is to combine multiple learners using different methods to obtain better fitting performance and smaller errors than a single model through advantage complementation among multiple models. In machine learning and data mining projects, various classifiers and models have their own advantages and disadvantages. Ensemble learning balances the advantages and disadvantages of each classifier to better complete classification and regression assignments. In recent years, many champions of machine learning competitions have used ensemble learning. Some mainstream internet companies, such as Tencent and Alibaba, have used ensemble learning in recommendation, search sorting, user behavior prediction, click-through rate prediction, product classification, and other businesses. Ensemble learning algorithms have achieved good results in these businesses. Existing research has used ensemble learning to predict $PM_{2.5}$ with reasonably accurate results [25]. However, the application of ensemble learning to visibility prediction has not been widely studied. As a result, the research in this paper focuses on developing a visibility prediction algorithm based on a fusion model to improve visibility prediction accuracy in the case of complex urban aerosol sources and frequent low-visibility scenes.

## 2. Data Source

The Key Laboratory of Atmospheric Optics of the Hefei Institute of Physical Sciences, Chinese Academy of Sciences, carried out a long-term meteorological and environmental monitoring test in Qingdao, Shandong Province (Institute of Marine Instrumentation, Shandong Academy of Sciences, 36.3°N, 120.18°E) from August 2019 to August 2020. The geographical information of the observation position is shown in Figure 1 (the observation position is denoted by the blue five-pointed star). The color in Figure 1 indicates the altitude of the observation position. Higher altitude is indicated by a redder color; lower altitude is indicated by a bluer color. Qingdao is located on the southeastern coast of Shandong Province. In addition, Qingdao has a typical pollution background, and its visibility is strongly correlated with pollutant parameters. It is also a typical northern coastal region of China. Research on visibility prediction in Qingdao can be extended to other northern coastal areas of China as well.

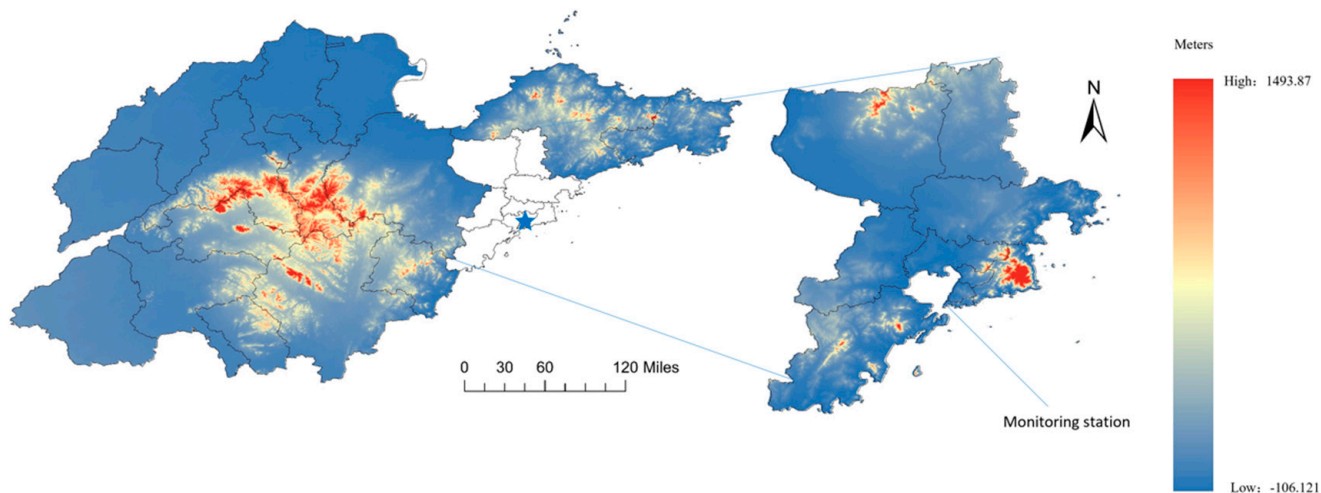

**Figure 1.** Locations of meteorological stations.

The data used in this paper included meteorological parameters, pollutant parameters, and visibility data measured in Qingdao, as listed in Table 1. Among these, meteorological data, such as average air pressure, air temperature, relative humidity, wind speed, and wind direction, were provided by ground weather stations with a time resolution of 5 s; $PM_{2.5}$, $PM_{10}$, $SO_2$, and other pollutant data were provided by air quality detectors with a time resolution of 1 h; and visibility data were provided by a scattering visibility meter with a time resolution of 1 min. The 6220 forward scattering visibility meter (Belfort Instrument, USA) was used to collect the visibility based on the optical parameter method. The scattering visibility meter uses the optical parameter method, which transmits the light into the sampler through an infrared LED transmitter, and the receiver collects the forward scattered light and calculates the extinction to obtain the atmospheric visibility.

**Table 1.** Description of raw parameters.

| Type | Parameter | Abbreviation | Unit | Description |
|---|---|---|---|---|
| Meteorological factors | Temperature | Temp | °C | Hourly average air temperature |
| | Humidity | Hum | % | Hourly average relative humidity |
| | Pressure | Pres | hPa | Hourly average barometric pressure |
| | Wind speed | WS | m/s | Hourly average wind speed |
| | Wind direction | WD | deg | Hourly average wind direction |
| Pollutant factors | $PM_{2.5}$ | $PM_{2.5}$ | $\mu g \cdot m^{-3}$ | Hourly average concentration of $PM_{2.5}$ |
| | $PM_{10}$ | $PM_{10}$ | $\mu g \cdot m^{-3}$ | Hourly average concentration of $PM_{10}$ |
| | $NO_2$ | $NO_2$ | $\mu g \cdot m^{-3}$ | Hourly average concentration of $NO_2$ |
| | $O_3$ | $O_3$ | $\mu g \cdot m^{-3}$ | Hourly average concentration of $O_3$ |
| | $SO_2$ | $SO_2$ | $\mu g \cdot m^{-3}$ | Hourly average concentration of $SO_2$ |
| | CO | CO | $\mu g \cdot m^{-3}$ | Hourly average concentration of CO |
| Visibility | Visibility | V | km | Hourly average visibility distance |

Data preprocessing consisted of data cleaning, data resampling, and data normalization on all experimental data. In data resampling, the three types of data above were resampled to a time resolution of 1 h. The meteorological and pollutant data were used as input for the model of visibility prediction. Model training and parameter optimization were carried out after constructing the training data sets to achieve the best model for visibility prediction.

### 3. Method

Previous research on visibility prediction based on machine learning methods has mostly used single models and has failed to fully consider the seasonal features of visibility created by differences in seasonal urban pollution sources. This study proposes a new visibility prediction method based on meteorological and pollutant data. The method adopts the stacking strategy to fuse the XGBoost and LightGBM models and introduces seasonal feature selection to improve the visibility prediction accuracy.

In the stacking method, the model output of the first layer is used as the input of the second layer model, and the result of the output of the second layer model is used as the final result. See Section 3.1 for details. In Section 3.2, a method and evaluation standard for the seasonal feature selection of training data set are established after analyzing the seasonal features of visibility and its possible causes. Evaluation criteria are also introduced in this section to evaluate the predictive performance of the models.

#### 3.1. Construction of the Fusion Model

Compared with single machine learning models, fusion models can achieve better fitting performance and smaller error through advantage complementation among multiple models.

The base learners used in this study were XGBoost and LightGBM models; then, the stacking strategy was used to fuse the two base learners. Both XGBoost and LightGBM models are improved algorithms based on the gradient boosting decision tree (GBDT) algorithm [28–30], and both have the characteristics of insensitivity to input requirements, low computational complexity, and good prediction effect. However, the two models show different advantages in different situations. For example, XGBoost performs better in the case of unbalanced data and less sample data, while LightGBM is faster in application, takes up less memory, and is less prone to overfitting. The basic learners and stacking fusion algorithm are detailed below.

#### 3.1.1. XGBoost

The eXtreme gradient boosting (XGBoost) algorithm was proposed by Tianqi Chen in 2015 [29]. Due to its efficient computation and support for custom loss functions, the algorithm is widely used in large-scale data processing and has become an important tool in the field of classification and prediction. The basic principle of XGBoost is the same as GBDT, but compared with GBDT, it has faster processing speed and higher accuracy, mainly reflected in the following aspects:

1. Regularization processing—XGBoost introduces regular terms to control the complexity of the tree in order to avoid overfitting and make the trained model simpler. In this way, its generalization performance is higher than that of GBDT.
2. Loss function optimization—second-order Taylor expansion is performed on the loss function, and the first-order derivative and second-order derivative information is used to determine the output result, which speeds up the training.
3. Strong flexibility—in addition to CART as a base learner, XGBoost also supports linear classifiers. Furthermore, XGBoost can customize the evaluation function, which is conducive to evaluating results from multiple performance metrics.
4. Improvement in the node division method—when seeking the optimal splitting point, XGBoost abandons traditional greedy algorithm segmentation and adopts the approximate greedy strategy algorithm to accelerate the leaf node splitting process and reduce the consumption of computing resources.

Compared with GBDT, XGBoost is superior in both accuracy and efficiency. This algorithm has been used for the prediction of visibility and $PM_{2.5}$, showing better performance than some other statistical and machine learning models [15,31,32].

### 3.1.2. LightGBM

The light gradient boosting machine (LightGBM) was proposed by Microsoft in 2017. On the basis of GBDT, this algorithm uses a histogram-based segmentation algorithm to replace the traditional pre-sort traversal algorithm [30]. The histogram algorithm discretizes continuous features into k discrete features and constructs a histogram with a width of k for statistical information. Using the histogram algorithm, we do not need to traverse the data, but only need to traverse k discrete feature nodes to find the best split node. Using histogram algorithm optimization not only reduces memory usage, but also reduces the computational cost.

In addition, LightGBM also adopts a leaf-wise algorithm with limited depth instead of the traditional level-wise decision tree growth algorithm. The traditional level-wise growth algorithm is convenient for calculating the split nodes of each layer in parallel, which improves the training speed. However, it also results in many unnecessary splits, because the node gain is too small. The leaf-wise growth strategy with limited maximum depth reduces unnecessary splits and improves prediction accuracy while avoiding the danger of overfitting.

Therefore, the LightGBM algorithm is superior to GBDT in terms of training speed and space efficiency. It can effectively prevent overfitting, making it more suitable for training with massive high-dimensional data. The algorithm has also been shown to be credible and efficient in $PM_{2.5}$ and visibility studies [16,33].

### 3.1.3. Stacking Fusion Model

The current mainstream ensemble learning methods are boosting, bagging, and stacking. Among them, bagging and boosting methods usually consider homogeneous weak learners. The former trains these weak learners in parallel independently and combines them according to an averaging method, while the latter is a highly automatic adaptive method that trains these weak learners sequentially and combines them according to a deterministic strategy. The stacking method in this research usually considers heterogeneous weak learners and combines them using a fusion model after training them in parallel. Then, the fusion model outputs a final prediction result according to the prediction results of different weak models [34]. The structure of this strategy is shown in Figure 2. The first stacking layer shows the working process of the base learners and the second stacking layer shows the working process of the fusion model. The stacking strategy uses five-fold cross-validation to avoid overfitting. The k of k-fold cross-validation is 2–10; k in this study is 5. The smaller the value of k, the smaller the amount of data available for modeling; the larger the value of k, the greater the training cost (training time). The specific steps are as follows:

1. Data set generation—firstly, the strategy generates a training set and a test set and divides the training set into five parts: train1, train2, train3, train4, and train5.
2. Base learner training—for each base learner (the first layer of stacking), train1, train2, train3, train4, and train5 are used in turn as the validation set, and the remaining four parts are used as the training set. Five-fold cross-validation is performed for model training and validation, and five copies of the validation ("predictions") are obtained. Then, predictions are made on the test set to obtain five predicted values.
3. Fusion model training—in this paper, a linear regression (LR) model was chosen as the fusion model (the second layer of stacking). The validations of the two base models are vertically overlapped (A1 and A2) as the train_x of the LR model and the corresponding true values are used as train_y of the LR model.
4. Fusion model prediction—the prediction values on the five test sets of base learners are averaged (B1 and B2) as the test_x of the LR model; then, using the trained LR model in Step 3 and test_x to make predictions, test_y is obtained as the final prediction result.

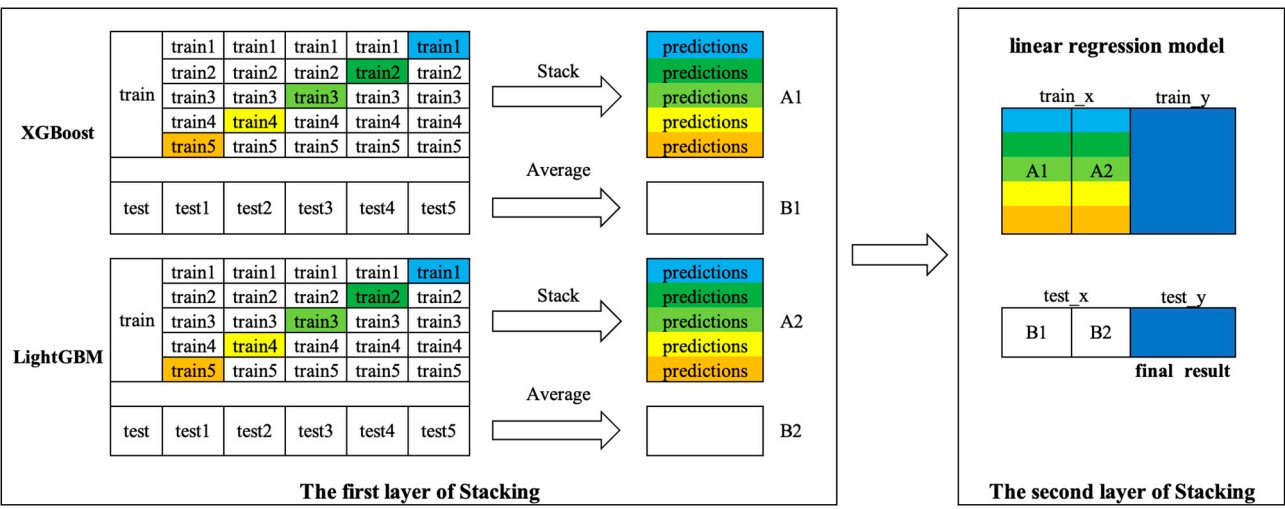

**Figure 2.** The model structure diagram of the VSFM.

*3.2. Feature Engineering*

In the work of visibility prediction, along with using the fusion model to improve visibility prediction accuracy, the proper use of feature engineering can also simplify the model, improve the training speed, and reduce errors in visibility prediction. In this section, monthly and hourly changes in visibility and the possible reasons for seasonal differences in visibility changes are analyzed. Then, two criteria for seasonal feature selection are proposed, and the results of seasonal feature selection are presented and discussed.

3.2.1. Seasonal Characteristics of Visibility in Qingdao

The visibility observation data in Qingdao were divided into four categories: spring (from March to May), summer (from June to August), autumn (from September to November), and winter (from December to February).

The hourly averaged visibility was calculated from 0:00 to 23:00 to derive its diurnal variations in each season. As shown in Figure 3, the visibility of the four seasons all showed a downward trend in the early morning, followed by an upward trend. The daily minimum values of the four seasons appeared at 6:00, 10:00, 6:00, and 7:00, respectively. This was mainly due to the increase in automobile exhaust emissions and air pollution accompanied by the early peak of people's travel. Among four seasons, the daily minimum of 2019_Winter was not significant compared to other seasons, and occurred later than other seasons. This may be related to the reduction in human activities during the city's closure because of COVID-19, resulting in a reduction in air pollution emissions, weakening the decreasing trend of V due to early peak travel. Generally, with the increase in temperature after sunrise and the decrease in relative humidity, thermal convection tended to be vigorous and visibility gradually improved. The daily maximum values of the four seasons appeared at 18:00, 15:00, 19:00–21:00, and 20:00, respectively. In the evening, as thermal convection conditions weakened and relative humidity increased, visibility deteriorated again.

In general, visibility in Qingdao in the four seasons was clearly differentiated. The visibility range was higher in the spring and autumn than in the winter and summer. The visibility in winter had the smallest change among seasons, ranging between 11 km and 15 km. This is due to heating in winter in Qingdao, and the overall air pollution was relatively serious. CO is mainly produced by automobile exhaust and heating. In Qingdao, the mean concentration of CO in winter was 0.92 $\mu g \cdot m^{-3}$, while in other seasons, it was 0.66 $\mu g \cdot m^{-3}$ (2019_Autumn), 0.59 $\mu g \cdot m^{-3}$ (2020_Spring), and 0.54 $\mu g \cdot m^{-3}$ (2020_Summer). In addition, CO concentrations in Qingdao during the winter were negatively correlated with visibility. Visibility in spring and autumn was relatively high. The analysis of influencing factors shows that the mean $PM_{2.5}/PM_{10}$ in spring and autumn was 15.32/68.21 $\mu g \cdot m^{-3}$ and 22.77/75.76 $\mu g \cdot m^{-3}$, while the mean $PM_{2.5}/PM_{10}$ in winter was 42.08/94.83 $\mu g \cdot m^{-3}$;

the mean relative humidity values in spring and autumn were 58.25% and 57.85%. This demonstrates that Qingdao has lower pollution levels in the spring and autumn, and the air is rather dry, resulting in quite good visibility. In summer, the concentration of pollutants is quite low (the mean $PM_{2.5}/PM_{10}$ is 14.63/29.45 µg·m$^{-3}$). However, the mean relative humidity is rather high (81.86%), and June and July are the most frequent months for sea fog in Qingdao [35], which has a great impact on the visibility. It is evident that there are seasonal differences in pollution sources and affecting factors in northern cities of China, which further supports the need for seasonal feature selection for visibility prediction in Qingdao.

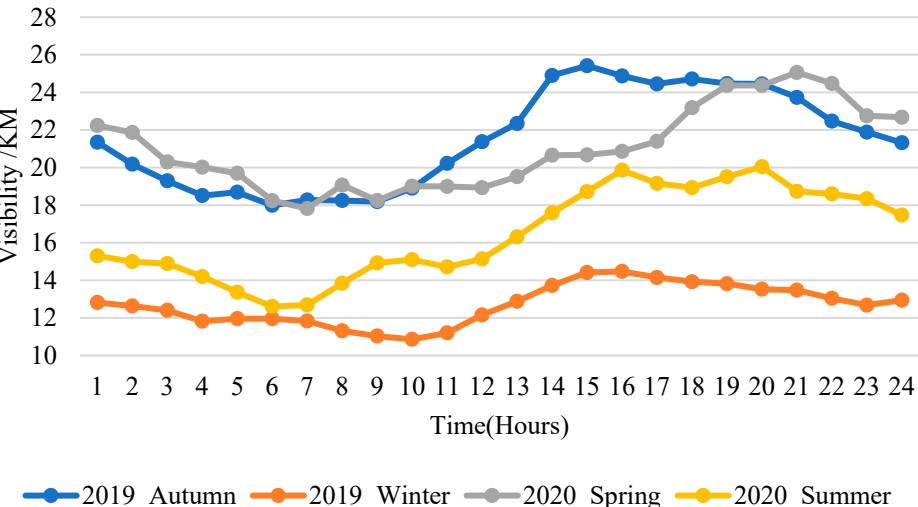

**Figure 3.** Diurnal variation of visibility in different seasons in Qingdao.

3.2.2. Seasonal Feature Selection

Feature engineering includes data cleaning, data resampling, data normalization, data transformation, and other data preprocessing and feature selection. In order to ensure prediction accuracy and simplify the model, it is necessary to select the most relevant features from the preprocessed features. Feature selection can identify and remove irrelevant and redundant features to prevent overfitting, and can also help reduce the feature dimensions and training time of the model.

From the previous section, we can see that there were seasonal differences in the visibility features in Qingdao, which resulted from differences in pollution sources in different seasons. Therefore, in the feature selection, we carried out a seasonal feature selection to clarify the different factors influencing visibility in different seasons. Similar work has been lacking in previous studies.

In this study, we used two criteria for feature selection. The first criterion used the "Filter" method to analyze the importance of each feature. The "Filter" method first assigned weights to the features of each dimension. Such weights represent the importance of the features in that dimension; then, they were ranked according to weight. To assign weights, we used the correlation coefficient (Corr):

$$Corr = \frac{\sum_{i=1}^{n}(x_i - \overline{x})(y_i - \overline{y})}{\sqrt{\sum_{i=1}^{n}(x_i - \overline{x})^2 \sum_{i=1}^{n}(y_i - \overline{y})^2}} \tag{1}$$

According to the first criterion, the correlation coefficient between visibility and meteorological and environmental parameters by season was scored. The results are shown in Table 2.

**Table 2.** Correlation coefficients between seasonal average visibility and various types of features in Qingdao.

| Season | Temp | Hum | Pres | WS | WD | $PM_{2.5}$ | $PM_{10}$ | $O_3$ | $NO_2$ | $SO_2$ | CO |
|---|---|---|---|---|---|---|---|---|---|---|---|
| 2019_Autumn | 0.07 | −0.36 | 0.13 | 0.03 | −0.19 | −0.66 | −0.51 | −0.35 | 0.05 | −0.30 | −0.59 |
| 2019_Winter | −0.20 | −0.45 | 0.37 | 0.18 | −0.15 | −0.60 | −0.56 | −0.43 | 0.19 | −0.27 | −0.65 |
| 2020_Spring | −0.12 | −0.60 | 0.45 | 0.01 | 0.09 | −0.62 | −0.35 | −0.29 | −0.04 | −0.21 | −0.62 |
| 2020_Summer | 0.10 | −0.39 | 0.00 | 0.04 | 0.05 | −0.44 | −0.40 | −0.15 | −0.07 | −0.10 | −0.34 |

The second criterion used a feature ranking based on the learning model. The feature ranking model was established for features screened by the first criterion. After the meteorological knowledge analysis and verification, the features were brought into the model verification, starting with the feature with the worst score. If it had a negative effect on the model training results, the feature was removed. The learning models used for feature ranking in this research were XGBoost and LightGBM. When calculating feature importance, the score was based on the number of times the feature appeared in the boosting tree, i.e., the times a certain feature was used as a split node in all generation trees.

Based on the calculation of the second criterion, the ranking results of the features of the two models are shown in Figure 4.

Comparing Figure 4a–d with Figure 4e–h, we found that the feature selection results of the two models were generally consistent, which is because both are based on gradient boosting tree algorithms. The factors affecting visibility in spring and summer are mainly humidity, $PM_{10}$, $PM_{2.5}$, and CO. Autumn and winter have more factors affecting visibility ($NO_2$ and $SO_2$) compared to spring and summer. Autumn has more factors affecting visibility (wind speed and wind direction) compared to winter.

In terms of factors affecting visibility common to the four seasons, there were several factors that always affected visibility in the ranking results of the two models, namely humidity, $PM_{10}$, $PM_{2.5}$, and CO. The aerosols in a coastal city such as Qingdao were found to have strong hygroscopicity, absorbing water vapor in humid conditions and decreasing V. Furthermore, high humidity causes moisture in the air to condense into tiny suspended water droplets, forming fog and strongly affecting visibility. $PM_{2.5}$ and $PM_{10}$ are important components of atmospheric particulate matter. The absorption and scattering effects of atmospheric particulate matter on sunlight can reduce visibility and haze. CO is mainly produced by heating and automobile exhausts. In heating and automobile exhaust emissions, CO is produced along with suspended solids, which is an important factor leading to reduced visibility in cities.

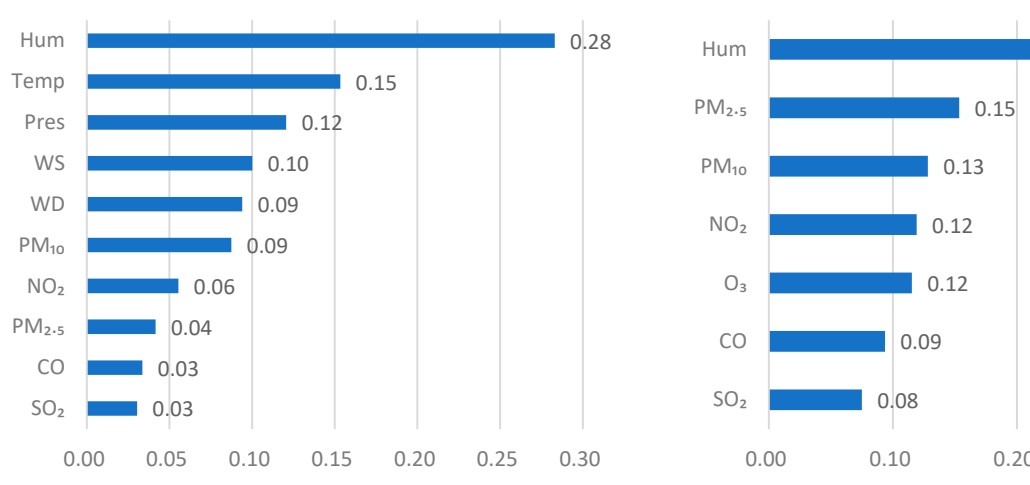

(**a**) XGBoost 2019_Autumn

(**b**) XGBoost 2019_Winter

**Figure 4.** *Cont.*

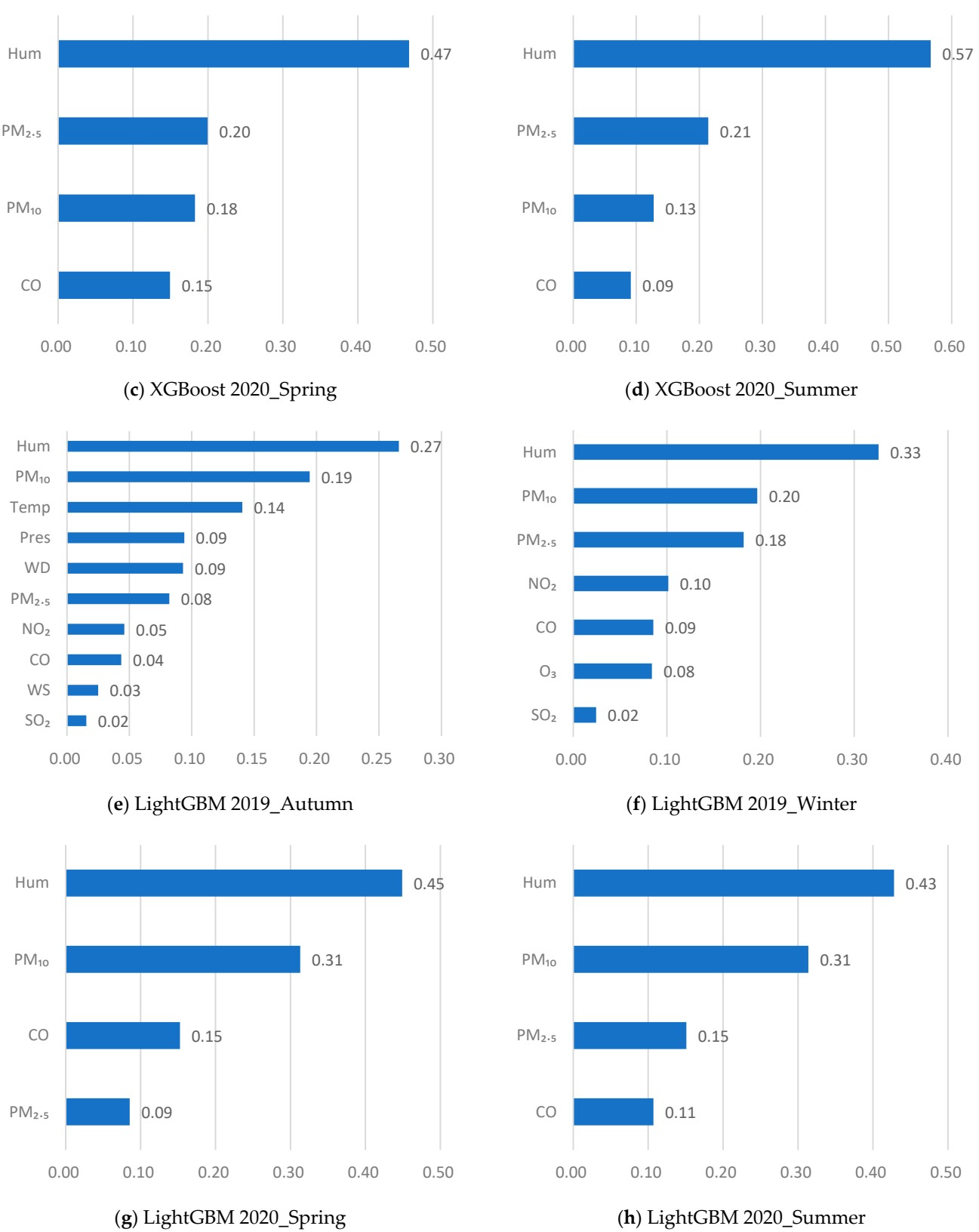

**Figure 4.** Feature selection results of XGBoost and LightGBM in each season. (**a**–**c**) and (**d**) are the feature selection results of XGBoost in 2019_Autumn, 2019_Winter, 2020_Spring, and 2020_Summer, respectively. (**e**–**h**) show the feature selection results of LightGBM in 2019_Autumn, 2019_Winter, 2020_Spring, and 2020_Summer, respectively.

By comparing the differences in the factors affecting visibility in the four seasons, we found that the $NO_2$ and $SO_2$ concentrations were included in the features of autumn and winter, but excluded in those of spring and summer. After analyzing the source data, we found that the mean $SO_2$ concentrations in autumn and winter (7.5 $\mu g \cdot m^{-3}$ and 8.9 $\mu g \cdot m^{-3}$, respectively) were higher than those in spring and summer (6.1 $\mu g \cdot m^{-3}$ and 5.0 $\mu g \cdot m^{-3}$, respectively). The $NO_2$ concentration has a similar result. The higher $SO_2$ and $NO_2$ concentrations in autumn and winter are closely related to urban heating in north China. Large amounts of atmospheric pollutants, such as $NO_x$ and $SO_2$, are produced during coal-burning for heating. $NO_x$ and $SO_2$ are the main pollutants, and $NO_x$ and hydrocarbons can easily generate secondary pollutants such as smog, affecting visibility under strong sunlight. Moreover, it can also be seen that the importance of wind speed and wind direction is higher in the autumn than in the winter. This may be because the heating time in the northern regions of China is earlier than in Qingdao, and pollution is transported to Qingdao by the north wind in the autumn. In winter, the local emissions from heating dominate and the weather is calm in Qingdao. Therefore, the impact of wind speed and wind direction on visibility is weak in the winter.

Furthermore, there were some differences in the specific feature rankings between the two models. This was because XGBoost adopted a level-wise splitting method, while LightGBM adopted a leaf-wise splitting method, resulting in differences in the ordering of features. This will be discussed in Section 5 in combination with the prediction results.

### 3.3. Performance Metrics

To quantitatively evaluate predictive effect, the root mean square error (RMSE), mean absolute error (MAE), and correlation coefficient (CC) were used for evaluation purposes. The RMSE and MAE were used to evaluate the degree of error, which can reflect the predicted extreme value and error range, and the CC was used to evaluate the degree of correlation between the predicted value and the observation value.

$$RMSE = \sqrt{\frac{1}{N} \sum_{i=1}^{N} (P_i - R_i)^2} \tag{2}$$

$$MAE = \frac{1}{N} \sum_{i=1}^{N} |P_i - R_i| \tag{3}$$

$$CC = \frac{\sum_{i=1}^{N} (P_i - \overline{P})(R_i - \overline{R})}{\sqrt{\sum_{i=1}^{N} (P_i - P)^2 \sum_{i=1}^{N} (R_i - \overline{R})^2}} \tag{4}$$

Here, "$N$" is the total number of predicted samples, "$P$" is the prediction result of the $i$th sample, and "$R$" is the actual value of the $i$th sample.

## 4. Results

Hourly meteorological and environmental parameter observation data in Qingdao from August 2019 to August 2020 were used to train, validate, and test the models. The performance of the VSFM was evaluated and compared with several existing numerical prediction methods, including XGBoost, LightGBM, SVM, MLR, and RF. For each set of seasonal data, we used the data set of the last 10 days as the test set and the data of the last 10–20 days as the validation set, while the rest of the data were used as the training set. This data division was performed to facilitate the presentation of the results. We also conducted experiments with randomly selected training, validation, and test sets; the experimental results did not differ significantly.

### 4.1. Comparison and Analysis of the Performance Metrics of Each Model

The performance metrics of the prediction results of each model were analyzed. The calculation results of the MAE, RMSE, and CC are shown in Table 3. In Table 3, the results

with and without feature selection are also presented to highlight the effect of the feature selection method proposed in this paper. The effect of feature selections was similar for the four seasons. In this paper, 2020_Summer was taken as an example to analyze the effect of feature selection. The experiments with feature selection used the common feature factors extracted in Figure 4 as training data for seasonal visibility prediction. The experiments without feature selection used all data as training data, including all meteorological and pollutant parameters.

**Table 3.** Error comparison of models in different seasons, where the MAE and RMSE are in km.

| | | Feature Selection | | | | All Features |
| | | 2019_Autumn | 2019_Winter | 2020_Spring | 2020_Summer | 2020_Summer |
|---|---|---|---|---|---|---|
| VSFM (ours) | MAE | 3.18 | 2.26 | 4.74 | 3.45 | 4.43 |
| | RMSE | 4.64 | 3.82 | 6.67 | 5.84 | 6.58 |
| | CC | 0.93 | 0.93 | 0.90 | 0.88 | 0.85 |
| XGB | MAE | 3.19 | 3.64 | 6.04 | 3.57 | 4.58 |
| | RMSE | 4.93 | 4.85 | 8.43 | 6.17 | 6.86 |
| | CC | 0.92 | 0.91 | 0.86 | 0.87 | 0.82 |
| LGBM | MAE | 3.65 | 2.52 | 5.50 | 3.73 | 4.93 |
| | RMSE | 5.01 | 4.11 | 7.48 | 6.43 | 7.17 |
| | CC | 0.91 | 0.91 | 0.87 | 0.84 | 0.83 |
| RF | MAE | 3.94 | 2.59 | 7.39 | 3.81 | 5.39 |
| | RMSE | 5.52 | 4.00 | 9.77 | 6.51 | 7.66 |
| | CC | 0.89 | 0.91 | 0.80 | 0.85 | 0.81 |
| SVM | MAE | 5.07 | 3.57 | 7.61 | 5.06 | 5.26 |
| | RMSE | 6.94 | 5.05 | 10.13 | 7.19 | 7.17 |
| | CC | 0.84 | 0.88 | 0.82 | 0.83 | 0.81 |
| MLR | MAE | 7.69 | 4.91 | 10.76 | 5.78 | 7.02 |
| | RMSE | 9.81 | 6.03 | 13.15 | 8.48 | 9.01 |
| | CC | 0.84 | 0.83 | 0.85 | 0.73 | 0.71 |

Comparing the prediction effects of the first three models and the latter three models, it can be seen that the VSFM, XGBoost, and LightGBM had better prediction effects in all seasons than RF, SVM, and MLR. In 2020_Spring, the prediction results of the VSFM, XGBoost, and LightGBM generally had significant advantages over the prediction results of RF, SVM, and MLR. This shows that the improved XGBoost and LightGBM based on GBDT are more suitable for visibility prediction than RF, SVM, and MLR, which is also the reason that XGBoost and LightGBM were chosen as the base learners in our VSFM.

Compared with XGBoost and LightGBM, the VSFM decreased RMSE by 0.0089–1.3795 km and decreased the MAE by 0.2819–1.7646 km, and increased the CC by 0.91–4.28% and the TS score by 0.0269–0.4573. This shows that the VSFM can effectively improve the accuracy of visibility prediction when the basic learner itself has a good prediction effect.

Comparing the prediction results with and without feature selection for 2020_Summer, it can be seen that the prediction effect of each model was improved to varying degrees after feature selection. The RMSE of the VSFM was decreased by 0.9834 km, the MAE was decreased by 0.7459 km, and the CC was increased by 2.99% after feature selection. In the performance metrics of those five models, the CC improvement in the XGBoost model was the most obvious, increasing from 82.46% to 86.95%.

Table 3 shows that the VSFM has superior performance to the other models in all four seasons, especially in 2020_Spring. Figure 5 shows the trend of the hourly visibility prediction curve for the six models in 2020_Spring. As shown in Figure 5, all the models underestimated the V under very clear conditions (V > 40 km). Some possible reasons are as follows. (1) Machine learning requires a large amount of historical data for accurate visibility prediction, and the amount of data with V > 40 km in the training set is small, meaning that it cannot meet the needs of model training and accurate prediction in machine learning. (2) The visibility observations in this study were obtained from a scattering visibility meter based on the optical parameter method. The optical parameter method (scattering method) is mainly used to calculate visibility by inverting the relationship between the atmospheric aerosol extinction coefficient and atmospheric visibility. This means that the higher the visibility, the weaker the scatterings and the larger the measurement error will be, thereby reducing the credibility of the data. Therefore, the accuracy of the prediction may be reduced. However, when V < 40 km, the prediction performance of the models was generally consistent and the prediction curve of the fusion model proposed in this paper was closest to the observation curve among models.

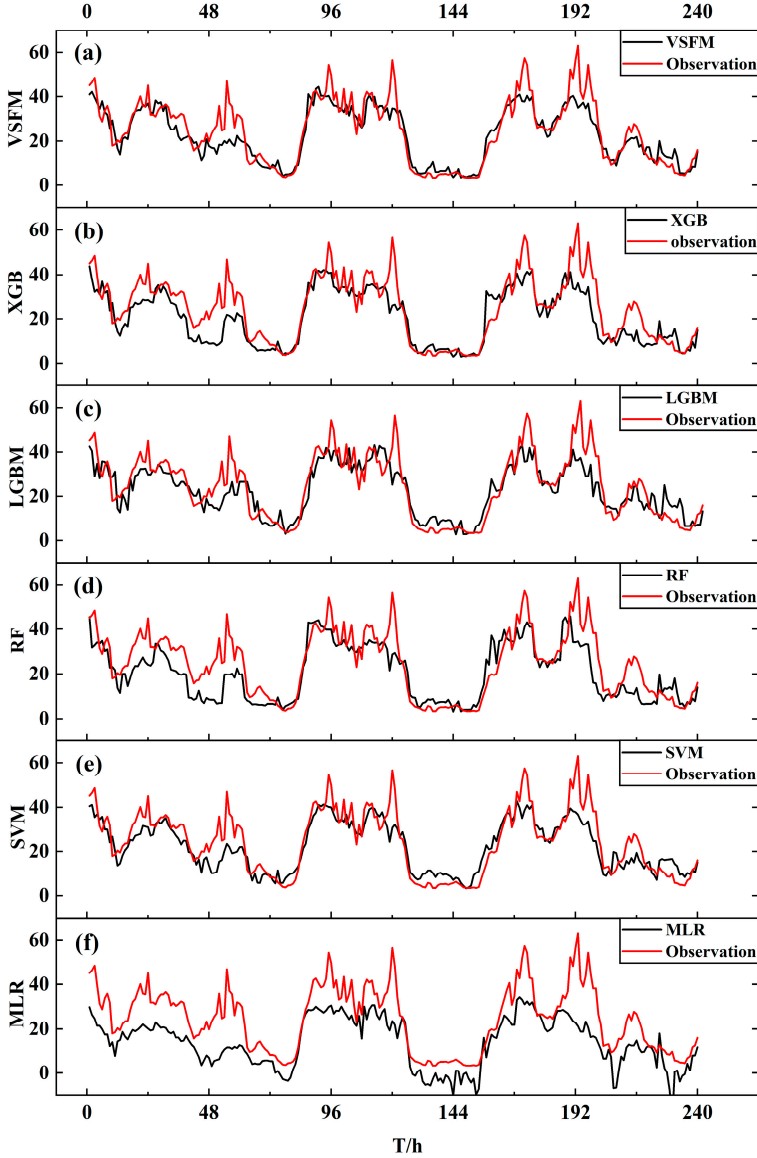

**Figure 5.** The prediction effect of each model in 2020_Spring. (**a–f**) show the prediction effects of the VSFM, XGBoost, LightGBM, RF, SVM, and MLR, respectively.

### 4.2. Visibility Classification Performance

Performance was further evaluated under different haze level conditions. The TS score commonly used in weather forecast studies [17] was adopted to evaluate the forecast accuracy of the models, defined as follows:

$$TS = \frac{n_a}{n_a + n_b + n_c} \tag{5}$$

where $n_a$ is the number of correct predictions (when the prediction results are in the classification and the actual results are also in the classification, a correct prediction is recorded), $n_b$ is the number of empty reports (when the prediction results are in the classification and the actual results are not in the classification, an empty report is recorded), and $n_c$ is the number of missing reports (when the forecast results are not in the classification and the actual results are in the classification, a missing report is recorded). In the calculation of the TS score, visibility is classified into the following four levels by reference to the haze level standard [21], as shown in Table 4.

**Table 4.** Description of visibility classification.

| Observations | Class | Rating |
|---|---|---|
| $0 < V < 2$ | I | Visibility is poor. |
| $2 < V < 5$ | II | Visibility is relatively bad. |
| $5 < V < 10$ | III | Visibility is relatively good. |
| $V > 10$ | IV | Visibility is excellent. |

The TS scores of models under different haze levels in different seasons are shown in Figure 6. Note that there were no Class I TS scores in 2019_Autumn and 2020_Spring due to relatively clean conditions in these two seasons. It can be seen from Figure 6 that the VSFM proposed in this paper performed better (a higher TS score) than the other models in the visibility classification test, especially under the extremely low-visibility conditions (Class I in 2019_Winter and 2020_Summer). In order to further study the cause of the low visibility in these two seasons, the mean meteorological parameters and pollutant parameters for Class I visibility in 2019_Winter and 2020_Summer were calculated and are presented in Table 5. As shown in Table 5, the extremely low visibility in 2019_Winter was mainly caused by $PM_{2.5}$ and $PM_{10}$, while the extremely low visibility in 2020_Summer was mainly caused by fog, as indicated by the very high humidity.

**Table 5.** Statistics on the mean values of meteorological parameters and pollutant parameters for Class I visibility in 2019_Winter and 2020_Summer.

| Season | Vis | Temp | Hum | Pres | WS | WD | $PM_{2.5}$ | $PM_{10}$ | $NO_2$ | $O_3$ | $SO_2$ | CO |
|---|---|---|---|---|---|---|---|---|---|---|---|---|
| 2019_Winter | 1.30 | 4.54 | 77.70 | 1022.26 | 2.89 | 242.87 | 85.41 | 185.87 | 55.59 | 40.76 | 9.48 | 1.59 |
| 2020_Summer | 1.07 | 21.85 | 89.15 | 1006.48 | 2.83 | 140.13 | 31.20 | 51.45 | 21.56 | 113.28 | 4.84 | 0.77 |

Table 6 summarizes the mean TS scores in all seasons for different models. As shown in Table 6, the VSFM performed better than the other models for four classifications in the total test set, especially the low-visibility classification. As can be seen from the Class I results, the highest TS score achieved by the single models was 0.2692, smaller than that of the VSFM. This is due to the fact that the VSFM utilizes the complementary advantages of multiple models, as described in Section 3.1. Therefore, the VSFM can reduce the number of empty reports $n_b$ (the $n_b$ values of the VSFM, XGBoost, and LightGBM for Class I were 4, 16, and 30) and missing reports $n_c$ (the $n_c$ values of the VSFM, XGBoost, and LightGBM for Class I were 2, 3, and 2) while maintaining the number of correct predictions $n_a$ (the $n_a$ values of VSFM, XGBoost, and LightGBM for Class I were 6, 5, and 6).

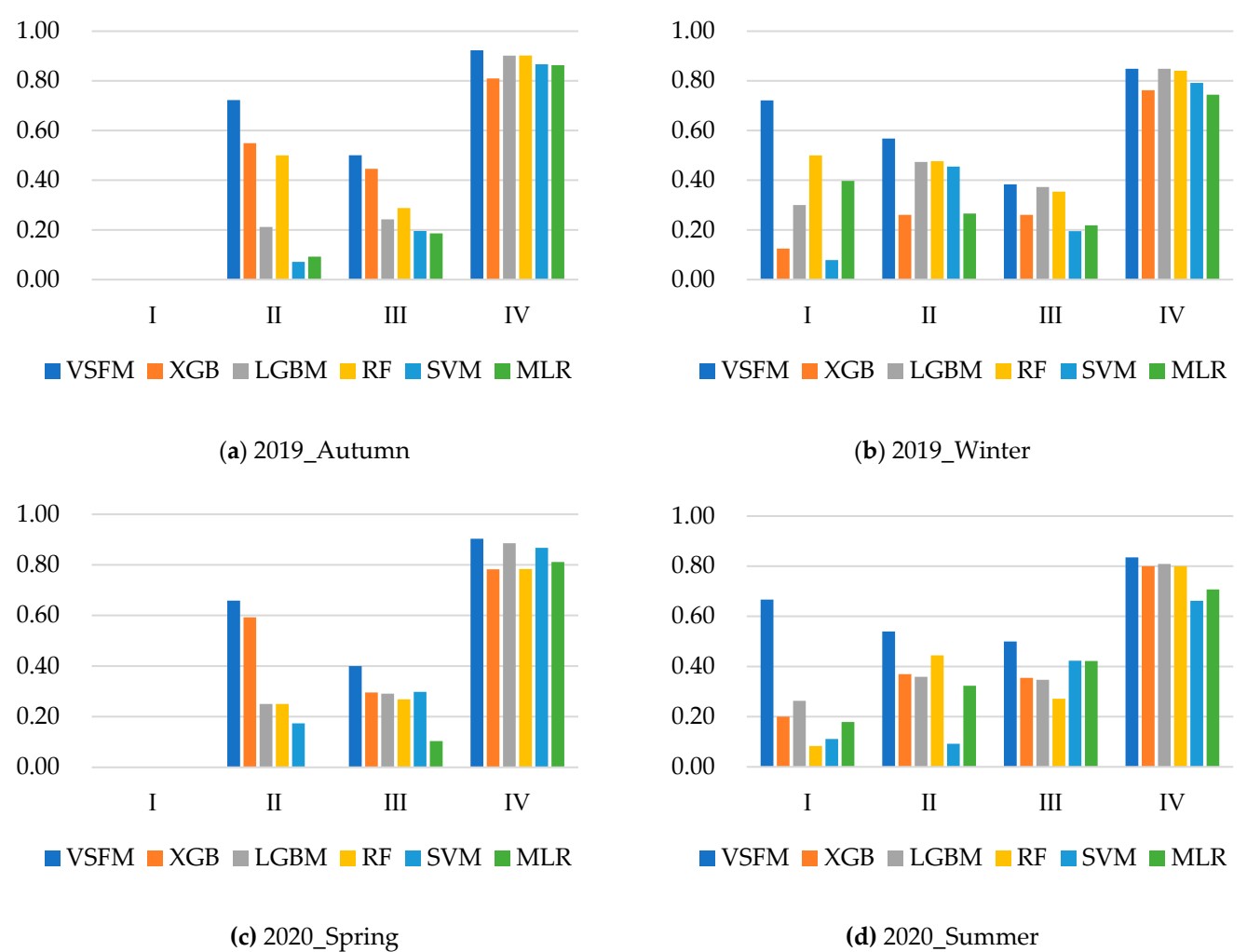

**Figure 6.** TS scores of several models. (**a**–**d**) are the TS score histograms of 2019_Autumn, 2019_Winter, 2020_Spring, and 2020_Summer, respectively.

**Table 6.** TS score performance in all seasons for different models.

| | I | II | III | IV |
|---|---|---|---|---|
| VSFM (ours) | 0.5000 | 0.5691 | 0.4286 | 0.8969 |
| XGB | 0.0781 | 0.3488 | 0.3055 | 0.8132 |
| LGBM | 0.2400 | 0.3373 | 0.3237 | 0.8700 |
| RF | 0.2692 | 0.4373 | 0.3000 | 0.8345 |
| SVM | 0.0427 | 0.2207 | 0.3007 | 0.8070 |
| MLR | 0.2261 | 0.1991 | 0.2552 | 0.7890 |

*4.3. Prediction Effect of the VSFM in Each Season*

Combining the above performance metrics, it can be seen that the VSFM proposed in this paper was superior to the other models in each performance metric. Figure 7 shows the predictive effect of the VSFM in each season. The VSFM had the best predictive effect in 2019_Winter, consistent with the results in Table 3. This may be because the visibility in Qingdao in winter is generally lower than 40 km due to heavy pollution as compared to other seasons. The test sets for other seasons all contained visibility data above 40 km to varying degrees. As mentioned in Section 4.1, due to the insufficient amount of data and the lack of confidence in the measurement instruments, it was difficult for the model to predict the peaks of V when the visibility was greater than 40 km. While it was challenging for the VSFM to predict the peak of V in very clean conditions (V > 40 km), the model could

predict the trends of V (e.g., 72–96 h in (a) and 168–192 h in (d) in Figure 7). Furthermore, there was no significant difference in visibility between 40 km and 60 km, and the effect of the very clean conditions made little difference to most people. Compared with the very clean conditions, extremely low-visibility conditions (Class I) were found to have a greater impact on people, and this paper achieved greatly improved prediction accuracy under extremely low-visibility conditions.

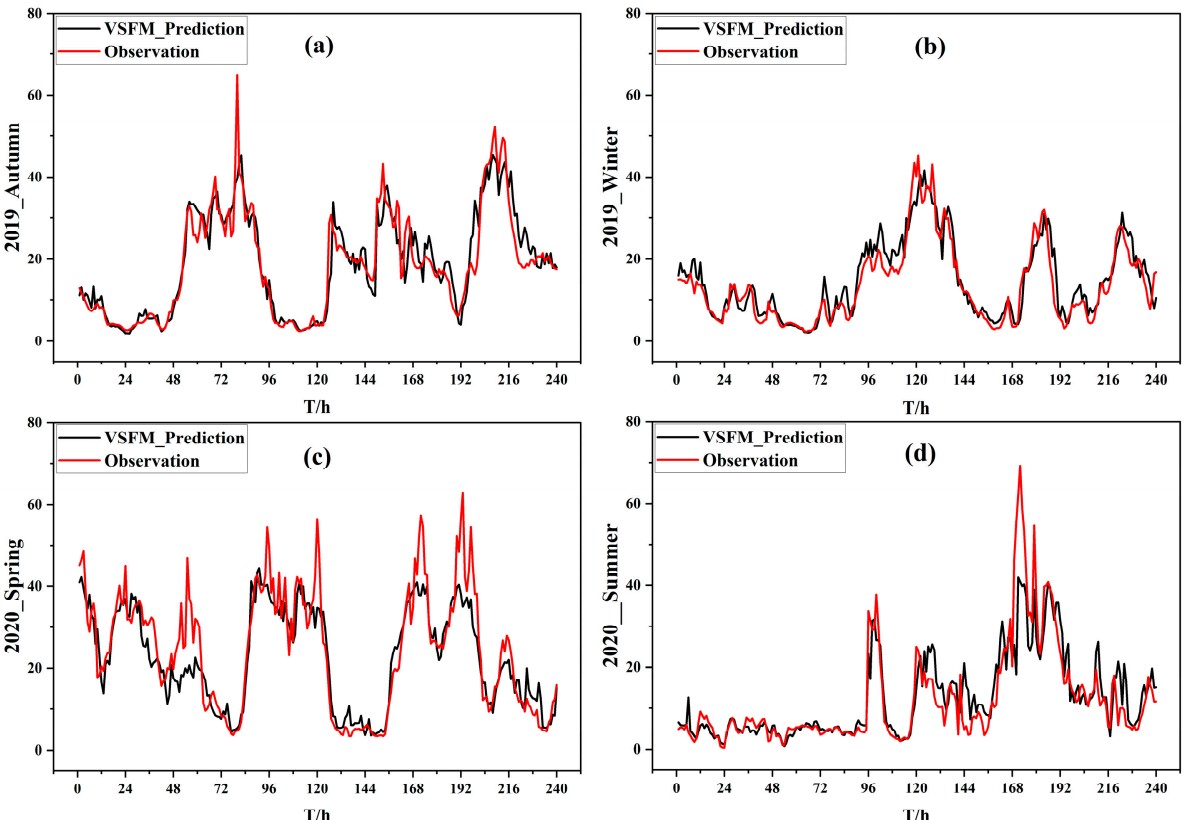

**Figure 7.** The prediction effect of the VSFM model in each season. (**a–d**) show the predicted values of the VSFM model versus the observed values for 2019_Autumn, 2019_Winter, 2020_Spring, and 2020_Summer, respectively.

## 5. Discussion

Previous studies on visibility prediction have often used a single machine learning model and historical meteorological data to determine the relationship between visibility and other observations, and have produced relatively good results [15–27]. However, there is still room for improvement in the prediction of low-visibility events that strongly affect the production and life of urban residents. This paper was mainly devoted to improving the accuracy of visibility prediction in terms of the model, visibility-influencing factors, and feature engineering.

In terms of the model, a stacking strategy was used to fuse two base learners based on machine learning to improve the visibility prediction effect. In terms of the selection of base learners, according to the study of Zhang et al. [27], Tang et al. [15], and Yu et al. [16], the XGBoost and LightGBM models were the most advanced regression algorithms and had excellent performance in predicting visibility. Therefore, XGBoost and LightGBM were selected as the base learners of the fusion model to carry out the visibility prediction task. In terms of factors affecting visibility, in addition to meteorological parameters, visibility is also affected by pollutants, aerosol chemical components, and other factors. Especially in China, the sources of urban pollution are extremely complex. Therefore, in this study, pollutant parameters were also used as training data for visibility prediction research

to improve visibility prediction accuracy. In feature engineering, this study found that there was a strong correlation between visibility changes and seasons, so we innovatively selected seasonal features for visibility. The prediction results of the model were compared with those of a single model. Results show that the VSFM's prediction performance was significantly better than that of any single machine learning method. The CC of the VSFM seasonal prediction results reached 92.96%, which is better than previous visibility prediction studies [15–24].

Although this study successfully applied the fusion model method to predict visibility in Qingdao, certain limitations should be considered. First, as described in Section 4.1, data with low reliability due to measurements by the visibility meter for visibilities greater than 40 km affected the final prediction accuracy. Another issue is that model fusion based on machine learning showed different prediction effects when applied to different seasons in Qingdao. One possible reason for the different seasonal behaviors of the VSFM is the predictive ability of base learners. Due to the different splitting methods of XGBoost and LightGBM, there were some differences between their feature sorting results and prediction results. As shown in the 2019_Winter prediction results, the predictive effect of LightGBM was slightly better than that of XGBoost (the CC of LightGBM is 0.9108 and that of XGBoost is 0.9065). Visibility in winter was mainly affected by coarse particulate ($PM_{10}$), while in the other three seasons, it was mainly affected by fine particles ($PM_{2.5}$). According to the feature selection results, the importance of $PM_{2.5}$ was generally higher than that of $PM_{10}$ in the feature ranking of XGBoost, while the importance of $PM_{10}$ was generally higher than that of $PM_{2.5}$ in the feature ranking of LightGBM. Therefore, the different sensitivities to $PM_{2.5}$ and $PM_{10}$ of the two models resulted in different prediction performances in winter. From the overall results of the three seasons, the prediction results of XGBoost were slightly better than those of LightGBM. Moreover, the importance of pressure and wind direction in XGBoost feature ranking was higher than that of $PM_{10}$ in 2019_Autumn, while the importance of $PM_{10}$ in LightGBM feature ranking was higher than that of air pressure and wind direction for the same period. This may indicate that XGBoost is more sensitive to the influence of air pressure and wind direction than LightGBM. Visibility in autumn was affected by externally transmissible pollutants, which may be related to pressure and wind direction. This may be why XGBoost has a better predictive effect than LightGBM. The difference between the prediction results of XGBoost and LightGBM in spring and summer may be related to their sensitivity to different hyperparameters. Although the VSFM model fused the results of the two base learners and improved the final prediction results, it can still have limitations. It is possible to further improve the accuracy of visibility prediction by introducing other deep learning algorithms, constructing new ensemble models for prediction, and introducing more factors affecting visibility.

## 6. Conclusions

A VSFM model for visibility prediction based on meteorological and pollutant parameter data was established through the fusion of two machine learning methods (XGBoost and LightGBM). In feature engineering, we established two feature selection criteria and extracted the feature factors of each season for visibility training and prediction after several effective data pre-processing steps, including data resampling, data normalization, missing value processing, etc. In addition, we compared performance metrics, which included the MAE, RMSE, CC, and TS scores, between the VSFM and single models to verify the improvement in the prediction accuracy of the VSFM. Single models for comparison included XGBoost, LightGBM, RF, SVM, and MLR. The following conclusions were drawn:

1.  The range of seasonal visibility changes in Qingdao and possible causes were analyzed in relation to meteorological and pollutant parameters. It was evident that there were seasonal differences in pollution sources and affecting factors in northern cities of China. Visibility was mainly affected by heating in the winter. In the summer, high humidity and frequent sea fog had a major impact on visibility. In the autumn, visibility was mainly affected by transported pollutants. Visibility was quite good in

the spring due to the rather dry and clean air. The seasonal differences in pollution sources and factors affecting visibility support the need for seasonal feature selection and visibility prediction in Qingdao.

2.  Based on the seasonal characteristics of visibility in Qingdao, seasonal feature selection for visibility prediction was designed and implemented. Feature importance criteria based on the "Filter" method and feature ranking were constructed for seasonal feature selection, and different feature factors were found that strongly correlated with the different pollution sources in different seasons. Evaluation results showed that applying feature selection reduced the RMSE by 0.9834 km and the MAE by 0.7459 km, and increased the CC by 2.99%;

3.  Performance metrics of the VSFM, such as the RMSE, MAE, and CC, were evaluated. The results showed that the VSFM could effectively improve the accuracy of visibility prediction when the basic learners performed well. The VSFM reduced the RMSE by 0.0089–1.3795 km and MAE by 0.2819–1.7646 km, and improved the CC by 0.91–4.28% and the TS score by 0.0269–0.4573 when compared to XGBoost and LightGBM;

4.  The TS scores of the VSFM and other single models were compared. The results showed that the VSFM significantly improved prediction accuracy under different classifications of visibility. Its advantage was especially obvious in extremely low visibility (V < 2 km, Class I). Under class I conditions, the VSFM had a TS score of 0.5, while the other models had scores less than ~0.27.

Overall, using meteorological and pollutant parameters as factors affecting visibility for seasonal feature selection, the VSFM model proposed in this study was found to effectively improve visibility prediction accuracy, especially in low-visibility scenarios. The VSFM outperformed other single models in visibility prediction due to its complementary advantages. Seasonal feature selection and excellent basic learners played an important role in obtaining high accuracy in visibility prediction. Furthermore, in addition to XGBoost and LightGBM, other models may be introduced in follow-up work, and more influential factors can be considered to improve the prediction accuracy of the VSFM. The model can also be extended to other regions by utilizing more data from these regions to form a more generalized visibility prediction model.

**Author Contributions:** Conceptualization, M.Z. and T.L.; methodology, M.Z. and T.L.; software, M.Z. and F.W.; validation, M.Z., F.W. and K.Y.; formal analysis, X.M. and F.W.; investigation, X.M. and K.Y.; resources, M.Y.; data curation, M.Z. and M.Y.; writing—original draft preparation, M.Z.; writing—review and editing, M.Z. and T.L.; visualization, M.Z.; supervision, T.L.; S.C. and X.L.; project administration, M.Z. and T.L.; funding acquisition, T.L. All authors have read and agreed to the published version of the manuscript.

**Funding:** This research was funded by the National Natural Science Foundation of China (grant no. 41875041); the Anhui Provincial Natural Science Foundation (grant no. 2008085j19); and the Youth Fund Project of the Advanced Laser Technology Laboratory of Anhui Province (grant no. AHL 2021QN01).

**Data Availability Statement:** The data presented in this study are available on request from the corresponding author.

**Acknowledgments:** We are grateful to our colleagues at the Key Laboratory of Atmospheric Optics for their help and input, without which this study would not have been possible.

**Conflicts of Interest:** The authors declare no conflict of interest. The funders had no role in the design of the study; in the collection, analyses, or interpretation of the data; in the writing of the manuscript; or in the decision to publish the results.

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
