# Peer review of "Application of a Fusion Model Based on Machine Learning in Visibility Prediction"

_remotesensing, doi:10.3390/rs15051450_

Round 1

Reviewer 1 Report

This manuscript “Application of fusion model based on machine learning in visibility prediction” presents a new visibility prediction method based on the Stacking Fusion Model (VSFM). This paper is well organized and includes many experiments and analysis. Still some questions need to be solved.

Comments are as follows:

1.     The method section is a little weak. The author should describe more details of the proposed method. For example, how does the feature selection work in the propose method and how to use LR model to make predictions.

2.     In line 308-309, “P” and “R” should be “” and “”.

3.     The division of datasets is unreasonable. The paper used the data set of the last 10 days as the test set, the data of the last 10–20 days as the verification set, and the rest were used as the training set, which is lack of randomness.

4.     The paper compares the results of 2020_Summer with and without feature selection. However, the processing details of experiment without feature selection is not mentioned. The results of other seasons should also tested in the experiment.

5.     The authors should compare the proposed method with 1-2 recently published methods at least to make the conclusion more convincing.

6.     The method only trained the model on the dataset of Qingdao in 1 year. Can the model still work on other regions?

Reviewer 2 Report

This paper presents a machine learning method for visibility prediction using measurements of meteorological and pollution data. Clarity of presentation improvement is needed.

Abstract should be rewritten. Use atmospheric visibility and define all acronyms in abstract. A contribution is a Stacking Fusion model, which is not described in the abstract section at all, but mostly the data interpretation is introduced. Authors used machine learning method to interpret data.

Figure 1 is not explained, it is referenced by pointing out the Qingdao.

Structure of the paper should be reorganized. Section 3.1 should be explained in detail, also XGbost and LightBGM algorithms should be mentioned in the introduction section along the state of the art section.

A block diagram of improvements in section 3.1. is needed.

Why 5 training sets are needed?

Why did you use XGBoost and LightGBM to predict B1 and B2?

Why did you not use SVM or other classifier, fed it with the data and observe the output? This approach probably does not work because of nature of data. Usually, preprocessing is needed.

The section 3.1 is the novelty contribution and is not well presented.

Data are commented in detail and in my opinion well this part is well presented.

How did you obtain the visibility observation data or can you describe the equipment (camera,…)?

On page 9 Fig. 4 is not well referenced.

How can you conclude that NO2 and SO2 concentration have great impact on visibility from Fig. 4?

In table 3 the results are measured in km, please define in caption.

Why the model VSFM can not follow the data and have large spikes, as shown in Fig. 7.

Can you depict similar image as Fig. 7 for other methods, reported in tables 3.5.

The first part of discussion section can be moved to the state of the art.

Reviewer 3 Report

Predicting atmospheric visibility has been challenging given the limited model capability. This study applied machine learning techniques to investigate the applicability of machine learning models in reproducing visibility in Qingdao city. While the topic is interesting, the major weakness of this study is the lack of innovation. The authors simply adopt several machine learning models and conduct inter-comparison for these model outputs, which is not innovative at all. Moreover, this study is limited to the local scale as only visibility data acquired at Qingdao is used. How could the research community benefit from the insights from this study? Can we expect the broad robustness of these models when performing nationwide applications? Lastly, the quality of the figures in the manuscript is vague and unclear to read (particularly for Figure 4. The authors even didn't put the numbers of individual air pollutants as subscripts). The above-mentioned limitations make this paper unpublishable in its present form. Therefore, I recommend this paper be rejected.

Round 2

Reviewer 1 Report

The authors have carefully replied to all of my questions and suggestions. I believe the manuscript can be accepted for publication.

Reviewer 2 Report

Authors answered to all of my questions, therefore, paper can be accepted in present form.